# Seeking the Oxidation Mechanism of Debris in the Fretting Wear of Titanium Functionalized by Surface Laser Treatments

**María del Carmen Marco de Lucas** [1,*], **Franck Torrent** [1], **Gianni-Paolo Pillon** [1], **Pascal Berger** [2] **and Luc Lavisse** [1]

1 Laboratoire Interdisciplinaire Carnot de Bourgogne (ICB), UMR 6303 CNRS-Université Bourgogne, 9 Av. Alain Savary, BP 47 870, CEDEX, F-21078 Dijon, France; gianni-paolo.pillon@u-bourgogne.fr (G.-P.P.); luc.lavisse@u-bourgogne.fr (L.L.)
2 Université Paris-Saclay, CEA, CNRS, NIMBE, 91191 Gif-sur-Yvette, France; pascal.berger@cea.fr
* Correspondence: delucas@u-bourgogne.fr

**Abstract:** Surface laser treatment (SLT) using nanosecond IR lasers has been shown to improve the tribological behaviour of titanium. Here, we studied the fretting wear of SLT-functionalized pure titanium in a mixture of reactive gases $O_2$ (20 vol.%) + $N_2$ (80 vol.%). The contact geometry was a ball on a plane and the ball was made of bearing steel. The very small amplitude of relative displacement between reciprocating parts in fretting wear makes the evacuation of wear particles difficult. Moreover, the oxidation mechanism of the debris depends on the accessibility of the surrounding atmosphere to the tribological contact. This work focused in the analysis of debris generation and oxidation mechanisms, and sought to differentiate the role of oxygen forming part of the ambient $O_2$ + $N_2$ gas mixture from oxygen present in the surface layer of the SL-treated titanium. Before the fretting test, the surface of the commercially pure titanium plates was treated with a laser under a mixture of $O_2$ + $N_2$ gases with oxygen enriched in the $^{18}O$ isotope. Then, the fretting tests were performed in regular air containing natural oxygen. Micro-Raman spectroscopy and ion beam analysis (IBA) techniques were used to analyse the $TiO_2$ surface layers and fretting scars. Iron oxide particles were identified by Raman spectroscopy and IBA as the third body in the tribological contact. The spatial distribution of $^{18}O$, Ti, $^{16}O$ and Fe in the fretting scars was studied by IBA. The analysis showed that the areas containing high concentrations of Fe displayed also high concentrations of $^{16}O$, but smaller concentrations of $^{18}O$ and Ti. Therefore, it was concluded that tribological contact allows the oxidation of iron debris by its reaction with ambient air.

**Keywords:** laser irradiation; fretting wear; tribological contact; ion beam analysis; Raman spectroscopy





## 1. Introduction

Titanium and its alloys are widely used in the aeronautical, marine and chemical industries for their good mechanical properties, high resistance to corrosion especially at room temperature and low density [1]. However, the low hardness and wear resistance of titanium [2] strongly limits its use in mechanical applications where the operating conditions lead to friction and wear. This can be the case in assembled structures where small amplitude oscillatory movements due to vibrations give rise to surface damage by fretting wear.

Different titanium alloys and surface treatments have been investigated in order to improve the tribological properties of titanium [3–6]. Among them, surface laser (SL) treatments are a powerful tool for modifying the surface composition and tribological properties of metals and alloys. Nanosecond pulsed lasers treatments have shown their ability to insert light elements (oxygen, nitrogen and carbon) from ambient gas into the surface layer whose composition and microstructure depend on the irradiation conditions, atmosphere and thermal properties of the target [7–12]. For pure titanium, it has been

shown that SL treatments using an infrared Nd:YAG nanosecond pulsed laser can reduce the friction coefficient in fretting conditions [13,14].

The range of relative displacement amplitude between reciprocating parts is very small in the case of fretting wear [15]. The evacuation beyond the contact region of the wear particles is difficult. Therefore, this debris participate in the fretting wear as a *third body*, the generic name used to describe the material imposed between the first bodies or generated as a result of interaction between the first bodies [16]. Moreover, the debris can form an oxide film and adhere to the contact surface of the first bodies [17–19]. This third body separates the interacting first bodies, and its thickness can vary from a monolayer to several micrometres [20]. Further experimental data on the composition, structure and size of the debris and its influence on the wear rate are still needed to advance our understanding of the role of debris, as well as new models that take into account their participation in fretting wear. In the past few years, different strategies have been developed to simulate the fretting wear accounting for the evolution of the debris trapped in the interface. Arnaud et al. [21] reported a finite elements approach to simulate Ti-6Al-4V cylinder–plane fretting interface. They concluded that the effect of the debris layer is geometrical rather than rheological, acting as a contact pressure concentrator favouring wear depth extension rather than lateral wear extension.

The ability of environmental elements to penetrate inside the contact to react with titanium (Ti17) was studied by Mary et al. [22]. In fact, the fretting contact is considered isolated, and the penetration of the ambient gas inside the contact is difficult and limited. Fouvry et al. [23] reported a fretting wear analysis of a dry Ti-6Al-4V cylinder–plane contact, developing an approach combining friction power and contact oxygenation. Recently, Arnaud et al. [24] developed an extended friction-energy wear approach, taking into account the debris layer and adhesive wear. For this, they simulated a di-oxygen partial pressure at the interface using an advection–dispersion reaction approach.

In this work we studied the fretting behaviour of pure titanium plates after an infrared (IR) nanosecond pulsed laser treatment in air, inducing the formation of a titanium dioxide surface layer. Previous works showed the presence of debris in the fretting scars mainly composed of iron oxide particles, which revealed a transfer of matter from the steel counterpart [13]. However, the oxidation mechanism of the debris remained uncertain depending on the accessibility of the surrounding atmosphere to the tribological contact. Here, we addressed this issue and sought to differentiate the role of oxygen in the ambient gas mixture $O_2 + N_2$ from that of oxygen present in the surface layer of SL treated titanium. For this, SL treatments within an $^{18}O$-enriched mixture of gases $O_2 + N_2$ were performed previously to fretting tests in regular air. Laser treatments using natural $O_2$ were also performed for comparison.

Micro-phase distribution in the surface layers and fretting scars was mainly studied by micro-Raman spectroscopy. The presence of iron oxides in the fretting scars was easily confirmed in this way.

Ion beam analysis (IBA) techniques were used to analyse the elemental and isotopic composition of the surface layers and fretting scars. In particular, nuclear reaction analysis (NRA) confirmed the presence of $^{18}O$ as the dominant oxygen isotope in the surface layers obtained under an $^{18}O$-enriched $O_2 + N_2$ gas mixture. The spatial distribution of $^{18}O$ and $^{16}O$ isotopes in the surface layers and fretting scars was mapped by NRA. Particle-induced X-ray emission (PIXE) showed the material transfer from the fretting ball to the sample. The correlation between the spatial distribution of $^{16}O$ and iron showed the role of ambient oxygen in the oxidation process of the debris coming from the fretting ball used in the test.

## 2. Materials and Methods

### 2.1. Laser Treatments

Commercially pure titanium plates (10 mm × 10 mm × 1 mm) were used for surface laser treatments. They were mechanically polished with a diamond paste and then washed with ethanol.

The laser treatments were performed using an IR Nd:YAG laser emitting pulses with a duration of 40 ns at 1064 nm. The laser spot (about 50 μm in diameter) was moved over the sample surface with a constant velocity (40 mm·s$^{-1}$) to form parallel straight lines with an interline spacing of 10 μm [11]. The laser repetition rate was 10 kHz and the laser power was 10 W. These parameters lead to an irradiance value around $25 \times 10^{12}$ W·m$^{-2}$, similar of that used in previous works [12,14,25].

The laser treatments were performed in a laboratory-sized chamber under a mixture of reactive gases composed of 80 vol.% of $N_2$ and 20 vol.% of $O_2$ enriched with isotope $^{18}O$. The goal was to be able to differentiate the oxygen incorporated in the sample surface during the laser treatment from that incorporated from the ambient atmosphere during the fretting test in regular air. These samples treated are named SLT-18O. For comparison, the same laser treatment was also performed using regular air containing natural oxygen. These samples treated are named SLT-16O.

### 2.2. Fretting Tests

The tribometer used is based on a piezoelectric actuator APA 120 ML from Cedrat technologies. The contact geometry was a ball on a plane. The ball was made of bearing steel (100Cr6 steel, hardness HRC 60) with a diameter of 24 mm. The Ti plates were stuck to a steel bulk with glue. The ball sample was subjected to alternating movement with an amplitude $\delta = \pm 50$ μm and a frequency of 10 Hz. The normal force imposed on the ball was kept at $11 \pm 1$ N leading to a contact pressure of $450 \pm 15$ MPa, lower than the compressive strength of titanium (around 680 MPa). Fretting tests were conducted with 20,000 cycles. The measured values of the normal and tangential forces, as well as the displacement amplitude, were acquired and processed by a specific software [26]. The ball movement was perpendicular to that of the laser scanning in the irradiated samples.

### 2.3. Characterization Techniques

IBA is a powerful tool to study the elemental chemical composition of surfaces, layers and interfaces [27–29]. Here, the composition and distribution of light elements in the layers were analysed by NRA, allowing the quantitative analysis of oxygen and nitrogen without the influence of the chemical environment and minimising the influence of roughness [12]. Deuterons were used as the incident beam in all NRA experiments. Two beam energies were chosen, either 920 keV, optimal for oxygen quantification from $^{16}O(d,p_1)^{17}O$ nuclear reactions, or 1900 keV, for nitrogen signals, especially $^{14}N(d,\alpha_1)^{12}C$ nuclear reactions [12]. Note that the same element may produce more than one peak when several nuclear energy levels are involved. The atomic oxygen and nitrogen concentrations were determined using $SiO_2$ (bulk) and a TiN layer as standards. The accuracy of the method was about 2%. The insertion depths were determined taking into account the experimental lowest energy value for an NRA peak assigned to a nuclear reaction involving the selected element as described in [14].

The distribution of titanium and iron in the fretting scars was studied by PIXE, whereas NRA was used to map the $^{16}O$ and $^{18}O$ spatial distributions. In particular, the intensity of the $^{18}O(d,\alpha_i)^{16}N$ (for i = 0 to 3) nuclear reactions was used to map $^{18}O$, whereas the $^{16}O(d,p_1)^{17}O$ nuclear reaction peak was used for $^{16}O$ maps.

Micro-phase distribution in the surface layers and fretting scars was studied by micro-Raman spectroscopy using an InVia Renishaw setup. The spectra were obtained in a back-scattering configuration. The excitation wavelength was 532 nm and the excitation power focused on the sample was about 0.5 mW to avoid heating the samples.

The roughness of laser-treated surfaces was measured by vertical scanning interferometry using a Veeco Wyko NT9100 optical profiling system. The surface roughness, $R_a$, was about 0.8 μm. Optical microscopy images (Figure S1), as well as profilograms (Figure S2) of the untreated and laser-treated titanium plates are given in the Supplementary Material.

## 3. Results and Discussion

### 3.1. Surface Layer Composition and Structure

Figure 1 shows two Raman spectra obtained for surface layers formed by laser irradiation under a gas mixture containing naturally occurring $O_2$ and $^{18}O$-enriched $O_2$, respectively. Both spectra show the Raman fingerprint of $TiO_2$ crystallized in the rutile phase, displaying three active modes ($B_{1g}$, $E_g$ and $A_{1g}$) in the 100–700 $cm^{-1}$ range (a very weak mode, $B_{2g}$, appears at higher frequencies). Note that Raman peak positions shifted to lower frequencies for the layer obtained with $^{18}O$-enriched $O_2$. This result agrees with the higher mass of $^{18}O$ compared to $^{16}O$. Moreover, the different peak shift values agree with those reported by [30] for $Ti^{16}O_2$ and $Ti^{18}O_2$ crystallized in the rutile phase. This confirms that $^{18}O$ is the dominant isotope of oxygen in the $TiO_2$ surface layer obtained using $^{18}O$-enriched $O_2$.

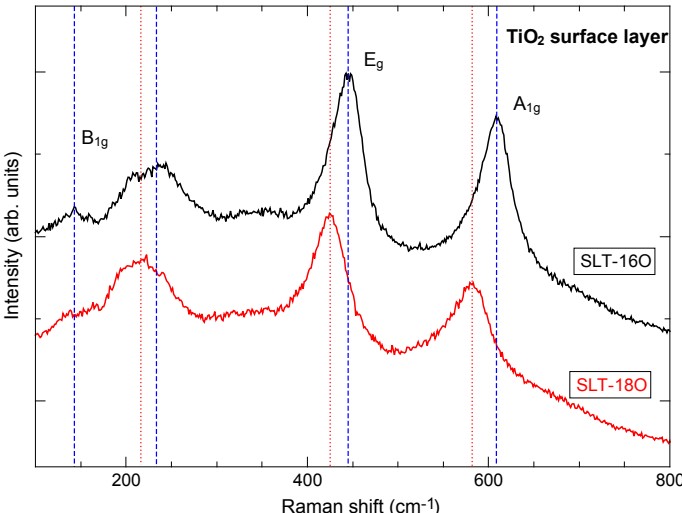

**Figure 1.** Raman spectra of the surface layers formed on top of Ti substrates by laser treatment using naturally occurring $O_2$ (sample SLT-16O) and $^{18}O$-enriched $O_2$ (sample SLT-18O), respectively. Dashed lines show the shift in the different Raman peaks as a function of the atmosphere used for surface laser treatment. Labels correspond to the active Raman modes of $TiO_2$ crystallized in the rutile phase.

NRA analysis was conducted to study the elemental composition of the surface layers. Experiments performed with the optimal conditions for nitrogen detection showed that the amount of nitrogen in the surface layer was negligible [14]. Figure 2 shows the NRA spectra obtained under optimal conditions for $^{16}O$ detection. For laser treatment using naturally occurring $O_2$, the oxygen concentration determined from the intensity of the $^{16}O(d,p_1)^{17}O$ NRA peaks was around 66 at.%, with titanium being around 34 at.%. Moreover, the oxygen insertion depth was estimated to about 0.6 μm [14]. For the surface layer obtained using $^{18}O$-enriched $O_2$, the NRA spectrum shows a lower intensity of all the peaks associated with nuclear reactions involving $^{16}O$, whereas several peaks assigned to nuclear reactions involving $^{18}O$ were identified, namely $^{18}O(d,p_i)^{19}O$ (for i = 0, 1) and $^{18}O(d,\alpha_i)^{16}N$ (for i = 0 to 3). The intensity of these peaks was strong enough that it was not necessary to analyse this sample using a proton incident beam as usually conducted for the optimal detection of $^{18}O$ by $^{18}O(p,\alpha_i)^{15}N$ nuclear reactions. This supports $^{18}O$ as the dominant isotope of oxygen in the $TiO_2$ surface layer obtained using $^{18}O$-enriched $O_2$ in agreement with the Raman spectroscopy results.

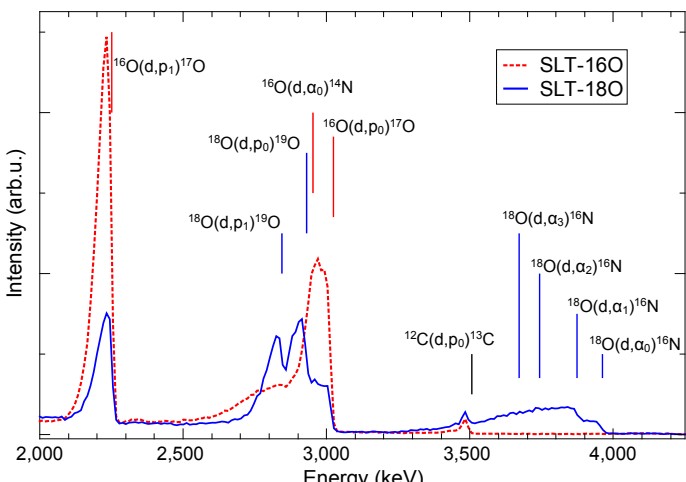

**Figure 2.** NRA spectra in the range of oxygen nuclear reactions obtained for the SLT-16O (dashed red line) and SLT-18O (solid blue line) samples. Spectra were obtained in experimental conditions optimized for oxygen detection: deuteron beam at 1900 keV, no Mylar foil. Peaks corresponding to the different nuclear reactions are indicated.

### 3.2. Fretting Results

Figure 3 shows the variation of the friction coefficient, $\mu$, as a function of the number of cycles, N, going up to 20,000 cycles, for the SLT-16O and SLT-18O samples. The variation in the friction coefficient of a polished titanium plate is also given for comparison. After a transition period, typically above 1000 cycles, the friction coefficient becomes almost constant for untreated Ti and SLT-16O samples. For SLT-18O a small drop in the friction coefficient was observed within the 10,000–15,000 cycle range. Above 15,000 cycles, the friction coefficient is similar for both the SLT-16O and SLT-18O layers with a value around 0.7, around 60% of the pure titanium sample, 1.2. Therefore, the surface laser treatment reduced the friction coefficient of pure titanium.

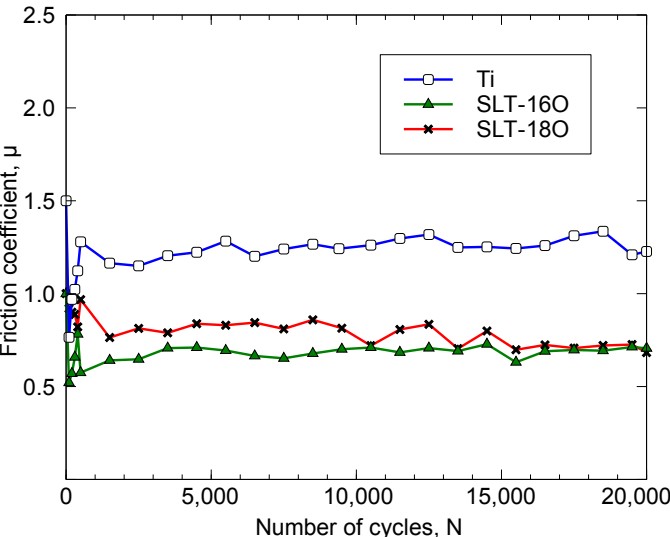

**Figure 3.** Evolution of the friction coefficient, $\mu$, with respect to the number of fretting cycles, N, obtained for the SLT-16O and SLT-18O samples. The curve $\mu(N)$ obtained for a sample of uncoated titanium is also shown for comparison.

### 3.3. Analysis of the Fretting Scars

Figure 4a shows the morphology of the fretting scar obtained after 20,000 cycles on the surface layer formed using $^{18}O$-enriched $O_2$. The scar is oval, about 1000 μm long and 900 μm wide. It shows some particles ejected outside the contact zone, whereas other

particles remain inside this zone. These particles, trapped in the contact area, play the role of the third body and contribute to kinematically adapting to the contact [16,31].

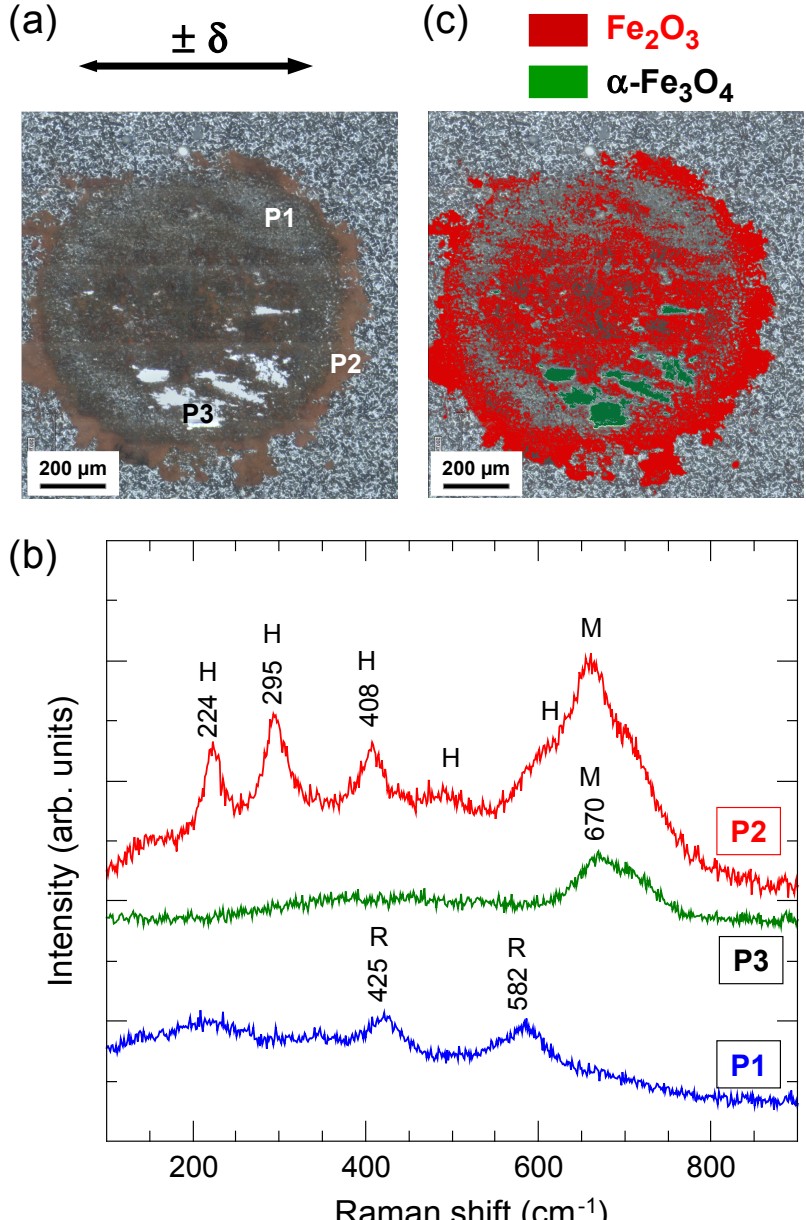

**Figure 4.** (**a**) Optical micrograph of the fretting scar formed after 20,000 cycles on the $TiO_2$ surface layer formed using $^{18}O$-enriched $O_2$ (SLT-18O sample); (**b**) Raman spectra obtained at points P1, P2 and P3 indicated in (**a**); (**c**) image of the fretting scar showing areas where the Raman spectra showed the presence of haematite ($\alpha$–$Fe_2O_3$, in red) or magnetite ($Fe_3O_4$, in green).

Around the scar, reddish particles suggest the presence of iron oxide in the haematite phase ($\alpha$–$Fe_2O_3$), caused by the oxidation of the iron fragments transferred from the friction ball to the surface layer.

The analysis of the scar was performed using Raman spectroscopy (Figure 4b,c). The fingerprint of the $TiO_2$ rutile phase was found in large areas of the scar (point P1) showing that the structure of the $TiO_2$ surface layer was not modified during the fretting test. Spectra obtained for the reddish particles in the periphery (point P2) and central areas of the scar confirmed the presence of iron oxide crystallized in the haematite phase ($\alpha$–$Fe_2O_3$) [32–34]. Raman spectra obtained in the shiny grey areas (point P3) in the centre of the scar can be identified as iron oxide crystallized in the magnetite phase ($Fe_3O_4$) [32–34].

The transfer of matter between the first bodies (friction ball and surface layer) was also studied by NRA and PIXE. Figure 5b–e show the maps of $^{18}O$, Ti, $^{16}O$ and Fe spatial distribution in the three regions of the fretting scar shown in Figure 5a. The maps obtained for a $200 \times 200$ $\mu m^2$ square region of the surface layer in an area away from the fretting scar are also shown for comparison in Figure 5b–d. Note, the homogeneous distribution of $^{18}O$, Ti and $^{16}O$ in the surface layer.

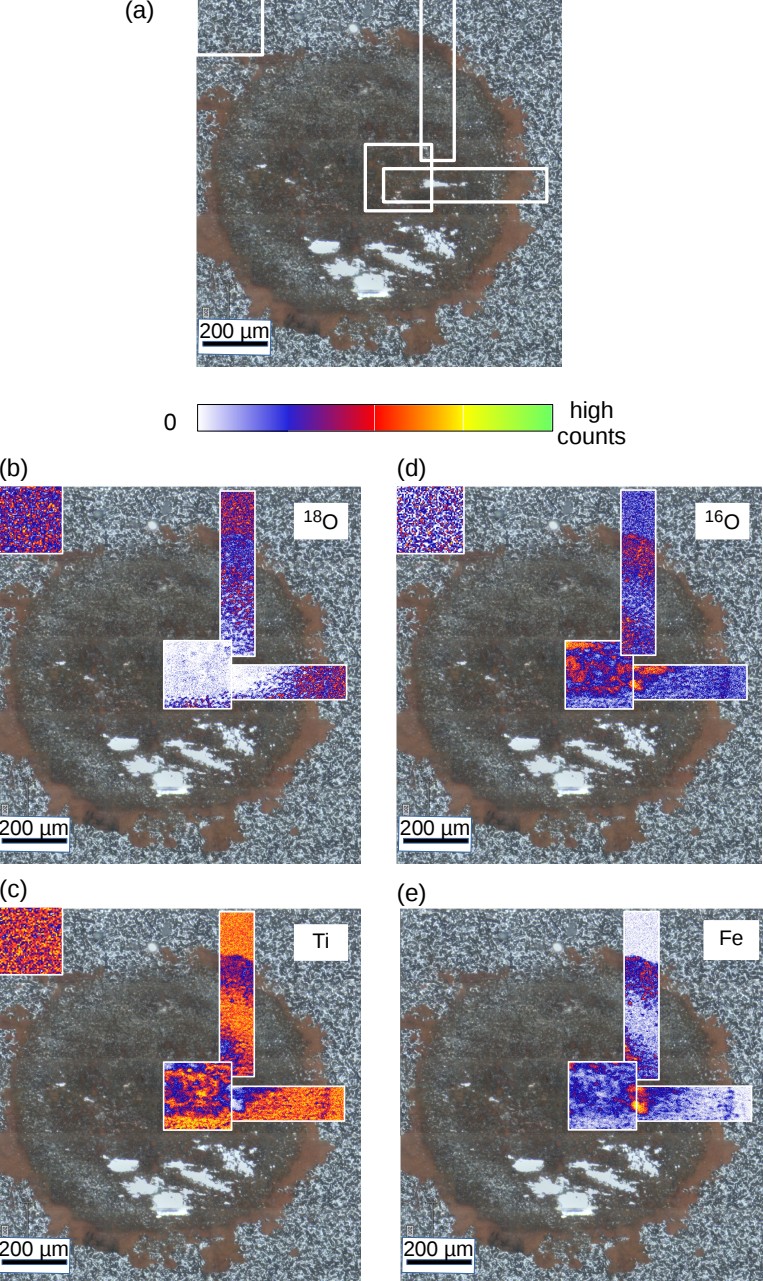

**Figure 5.** NRA and PIXE maps displaying the spatial distribution of $^{18}O$, Ti, $^{16}O$ and Fe in different areas of the fretting scar in Figure 4 for the SLT-18O sample, and in the surface layer away from the fretting scar. (**a**) Optical micrograph showing the analysed areas: square areas at the centre and outside of the fretting scar ($200 \times 200$ $\mu m^2$), rectangular areas perpendicular to the fretting direction ($100 \times 500$ $\mu m^2$), and parallel to the fretting direction ($500 \times 100$ $\mu m^2$). (**b,d**) Spatial distribution of $^{18}O$ and $^{16}O$, respectively, obtained by mapping the intensity of the $^{18}O(d,\alpha_i)^{16}N$ (for i = 0 to 3) and the $^{16}O(d,p_1)^{17}O$ nuclear reaction peaks in the areas shown in (**a**). (**c,e**) Spatial distribution of Ti and Fe, respectively, obtained by mapping the intensity of the corresponding PIXE signals.



In the scar, Figure 5b,c show the similar spatial distributions of $^{18}$O and Ti. The presence of iron in the fretting scar, and thus, the matter transfer from the friction ball to the sample surface is confirmed in Figure 5e. Moreover, the areas showing high concentrations of Fe also display high concentrations of $^{16}$O (Figure 5d), but smaller concentrations of $^{18}$O and Ti. Therefore, we conclude that iron debris from the friction ball are transferred to the sample surface and oxidized by reactions with the ambient air. The contribution of another oxidation mechanism involving the diffusion of oxygen ($^{18}$O) from the surface layer to the third body (iron fragments) seems negligible here.

This result differs form the conclusions of Mary et al. [22] who studied the tribological transformations occurring in Ti alloys under fretting in air at temperatures raging from ambient to 450 °C. Wear debris containing a high content of nitrogen and the formation of a $TiN_xO_y$ phase in the contact were shown [22]. However, the parameters of the tribological contact were very different to those in the present study. The contact geometry was a large-scale punch (14 mm$^2$ flat surface) on a plane to reproduce the blade–disk contact geometry. Thus, the possibility for environmental elements to penetrate inside the contact and react with the titanium was limited. Two mechanisms were proposed by Mary et al. [22] explaining the nitrogen penetration into the contact: oxygen depletion and mechanically favoured diffusion. In our work, the surface contact was quite smaller than in the work of Mary et al. [22]. The results show that ambient air can penetrate into the contact area and induce oxidation of iron debris.

The isotopic labelling using $^{18}$O-enriched oxygen during the laser treatment of the surface, combined with NRA to study the oxygen isotope spatial distribution in the fretting scars, proves to be an effective method to study the oxidation mechanism when different contributions are involved.

## 4. Conclusions

Surface laser treatments using nanosecond IR lasers have been shown to modify the behaviour of pure titanium under fretting wear conditions. This work focused on the analysis of debris generation and oxidation mechanisms in the fretting wear of SLT-functionalized pure titanium in a mixture of reactive gases $O_2$ (20 vol.%) + $N_2$ (80 vol.%). Furthermore, we sought to differentiate the role of oxygen in the ambient gas mixture $O_2$ + $N_2$ from oxygen present in the surface layer of the SL-treated titanium. The contact geometry was a ball on a plane and the ball was made of bearing steel.

The analysis of the debris by micro-Raman spectroscopy and IBA techniques (NRA and PIXE) has shown that iron fragments from the friction ball are transferred to the SLT-functionalized titanium surface and oxidized. They play the role of a third body in the tribological contact.

Laser treatments within an $^{18}$O-enriched mixture of gases $O_2$ + $N_2$ were performed previously to the fretting tests in air in order to investigate the accessibility of the surrounding atmosphere to the tribological contact. The analysis of the surface layers obtained showed the formation of $TiO_2$, where $^{18}$O was the dominant oxygen isotope. In the fretting scars, Raman spectra showed the presence of iron oxide particles crystallized in the haematite ($\alpha$–$Fe_2O_3$) and magnetite ($Fe_3O_4$) phases.

The spatial distribution of $^{18}$O, Ti, $^{16}$O and Fe in the fretting scars was studied by NRA and PIXE. This showed that areas showing a higher concentration of $^{18}$O also had a higher concentration of Ti. PIXE maps confirmed the presence of iron in the fretting scar. Moreover, the areas showing high concentrations of Fe also displayed higher concentrations of $^{16}$O, but smaller concentrations of $^{18}$O and Ti. Thus, an oxidation mechanism involving the diffusion of oxygen ($^{18}$O) from the surface layer to the third body (iron debris) can be neglected. It was concluded that the tribological contact allows the oxidation of iron debris by reaction with ambient air.

**Supplementary Materials:** The following supporting information can be downloaded at: https://www.mdpi.com/article/10.3390/coatings13061110/s1, Figure S1: Optical microscopy images of laser-treated titanium; Figure S2: Surface profilograms.

**Author Contributions:** Conceptualization, M.d.C.M.d.L. and L.L.; data curation, M.d.C.M.d.L., F.T. and L.L.; formal analysis, M.d.C.M.d.L., F.T., P.B. and L.L.; investigation, M.d.C.M.d.L., F.T., G.-P.P., P.B. and L.L.; methodology, M.d.C.M.d.L., F.T., G.-P.P., P.B. and L.L.; project administration, L.L.; validation, M.d.C.M.d.L., G.-P.P., P.B. and L.L.; visualization, M.d.C.M.d.L.; writing—original draft, M.d.C.M.d.L. and F.T.; writing—review and editing, M.d.C.M.d.L., P.B. and L.L. All authors have read and agreed to the published version of the manuscript.

**Funding:** This work was supported by the EIPHI Graduate School (contract ANR-17-EURE-0002).

**Institutional Review Board Statement:** Not applicable.

**Informed Consent Statement:** Not applicable.

**Data Availability Statement:** The raw/processed data required to reproduce these findings cannot be shared at this time as the data also forms part of an ongoing study.

**Acknowledgments:** The authors thank the Intense Lasers and Applications Center (CELIA)—UMR5107 CNRS–Université de Bordeaux for the provision of a chamber for the treatment of the samples in a controlled atmosphere and the Mechanical Resource Center of the ICB Laboratory for technical support.

**Conflicts of Interest:** The authors declare no conflicts of interest.

## Abbreviations

The following abbreviations are used in this manuscript:

| | |
|---|---|
| SL | Surface laser |
| SLT | Surface laser treatment |
| IBA | Ion beam analysis (IBA) |
| NRA | Nuclear reaction analysis |
| PIXE | Particle-induced X-ray emission |

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
