# Peer review of "Seeking the Oxidation Mechanism of Debris in the Fretting Wear of Titanium Functionalized by Surface Laser Treatments"

_coatings, doi:10.3390/coatings13061110_

Round 1
Reviewer 1 Report
The current level of development of laser systems and technologies provides new opportunities for fast and high-quality precision machining of material surfaces. The positioning accuracy and the small diameter of the focused laser beam make it possible to reach the micro- and nanometer scale. The convenience of controlling the parameters of laser systems, the simplicity and accuracy of pointing laser radiation into the treatment zone, and the reproducibility of the results make the process of laser structuring of surfaces and increasing wear resistance competitive. In the article, new scientific results useful for science and practice are obtained. But there are a number of questions:
1. What is the depth of the modified microstructure layer using nanosecond IR lasers?
2. Why did the authors not study the change in the microhardness of the modified microstructure layer after processing with nanosecond IR lasers?
3. What was the surface roughness after processing with nanosecond IR lasers?
4. I recommend the authors to add an analysis of the article https://link.springer.com/article/10.1007/s11668-014-9869-4, in the Introduction.
5. The structure of the article should be corrected to a generally accepted form. Figures must be inserted into the text before conclusions, etc.
Reviewer 2 Report
The manuscript presents the experimental results of surface laser treatments using nanosecond IR lasers on pure titanium under fretting wear conditions. The study focused on analyzing debris generation and oxidation mechanisms in the fretting wear of surface laser-treated (SLT) functionalized pure titanium in a mixture of reactive gases O2 (20 vol.%)+ N2 (80 vol.%). The objective was to differentiate the role of oxygen in the ambient gas mixture O2+N2 from that of oxygen present in the surface layer of SL-treated titanium. The contact geometry involved a ball on a plane, and the ball was made of bearing steel.
The experimental investigation demonstrated that the frictional coefficient was reduced due to the surface treatment for both SLT 16O and SLT 18O. The study's conclusion was based on empirical results obtained through experimental investigation. Although the study is interesting, the English language used in the manuscript is a bit challenging to follow. For instance, the abstract states, "Laser treatments under an 18O-enriched gas mixture were done previously to fretting tests in regular air," which makes it unclear whether the work is from the authors or literature, rendering the abstract difficult to understand.
To make the manuscript publishable, the following modifications need to be made:
Correct the referencing style. For example, Line 211 should read "Mary et al. [22]."
Explain the reason for using the symbol for a 40 ns pulse in Line 93.
Clarify why 18O-enriched O2 was chosen since it does not naturally occur, as requested in Line 106.
Remove the symbols in Line 110, as they were not used in any form.
Overall, the manuscript presents valuable experimental findings that can contribute to the development of effective surface laser treatments. With the necessary modifications, the study can be more easily understood and appreciated by readers in the field.
The grammatical accuracy of the English used in the manuscript is quite good. However, the writing style is not very coherent, making it challenging to comprehend the author's intended message.
Reviewer 3 Report
In general the paper is well-written and the topic of the paper is relevant to the readers of Coatings journal. The-state-of-the-art in the field is thoroughly presented in the Introduction, used methods are clearly described in the M&M, and obtained results are clearly presented both in Figures and the text. However, although Coatings accepts free format submission I believe that the paper would be improved if obtained results are discussed in separate section. Therefore, I would suggest to separate Results and Discussion into two sections.
Round 2
Reviewer 1 Report
The authors answered most of the questions, but I would like to further clarify the following:
1. During laser processing work, the surface quality may be outside the specified range of requirements for a number of reasons, namely:
- during treatment, boiling of the melt bath may occur, as a result, it is necessary to have an early warning system for emerging defects;
- during treatment, the roughness of the machined surface may change due to the general heating of the body of the part;
- roughness after processing depends on the initial state of the surface, which, for example, after machining is not always the same in different areas.
It would be nice to show surface profilograms before and after processing.
2. in the process of processing, a whole complex of physical and mechanical characteristics of the surface layer of the part is modified, which includes roughness, phase composition, microhardness, residual stresses. Did you manage to find physical and mechanical patterns of changes in these parameters?
3. The authors did not investigate microhardness, what then was the parameter used as a parameter for controlling the properties of the hardened layer ?
4. It would be nice to (visualize) show the surface before and after processing at various magnifications.
Author Response
"Please see the attachment."

Round 3
Reviewer 1 Report
Accept.